

# Genome-wide identification of C2H2 zinc-finger genes and their expression patterns under heat stress in tomato (*Solanum lycopersicum* L.)

Xin Hu, Lili Zhu, Yi Zhang, Li Xu, Na Li, Xingguo Zhang and Yu Pan

Key Laboratory of Horticulture Science for Southern Mountainous Regions, Ministry of Education, College of Horticulture and Landscape Architecture, Southwest University, Chongqing, China
Academy of Agricultural Sciences, Southwest University, Chongqing, China

## ABSTRACT

The C2H2 zinc finger protein (C2H2-ZFP) transcription factor family regulates the expression of a wide variety of genes in response to various developmental processes or abiotic stresses; however, these proteins have not yet been comprehensively analyzed in tomato (*Solanum lycopersicum*). In this study, a total of 104 *C2H2-ZFs* were identified in an uneven distribution across the entire tomato genome, and include seven segmental duplication events. Based on their phylogenetic relationships, these genes were clustered into nine distinct categories analogous to those in *Arabidopsis thaliana*. High similarities were found between the exon–intron structures and conserved motifs of the genes within each group. Correspondingly, the expression patterns of the *C2H2-ZF* genes indicated that they function in different tissues and at different developmental stages. Additionally, quantitative real-time PCR (qRT-PCR) results demonstrated that the expression levels of 34 selected *C2H2-ZFs* are changed dramatically among the roots, stems, and leaves at different time points of a heat stress treatment, suggesting that the C2H2-ZFPs are extensively involved in the heat stress response but have potentially varying roles. These results form the basis for the further molecular and functional analysis of the C2H2-ZFPs, especially for those members that significantly varied under heat treatment, which may be targeted to improve the heat tolerance of tomato and other Solanaceae species.

Corresponding author
Yu Pan, panyu1020@swu.edu.cn

## INTRODUCTION

The zinc finger proteins (ZFPs) are one of the largest protein families in plants. These proteins harbor a highly conserved domain, in which a zinc ion is surrounded by cysteine (Cys) and/or histidine (His) residues to stabilize their three-dimensional structure, comprising a two-stranded antiparallel beta sheets and a helix (*Liu et al., 2015*). Based on the number and location of these Cys and His residues, the ZFPs can be divided into ten classes: C2H2 (TFIIIA), C2HC (Retroviral nucleocapsid), C2HC5 (LIM domain), C2C2,

C3H (Nup 475), C3HC4 (RING finger), C4 (GATA-1), C4HC3 (Requium), C6 (GAL4), and C8 (Steroid-thyroid receptor) (*Berg & Shi, 1996*; *Liu et al., 2015*; *Michael & Chrisopher, 2003*). The members of these classes play crucial roles in plant growth and development, as well as in signal transduction and environmental stress responses (*Takatsuji, 1998*; *Stege et al., 2002*; *Le Gall et al., 2015*; *Yin et al., 2017*).

The C2H2-ZFPs account for a large proportion of the ZFPs and contain a characteristic motif, X2-Cys-X(2-4)-Cys-X12-His-X(3-5)-His (where X represents any amino acid) (*Takatsuji, 1999*), which has been observed in many eukaryotes (*Fedotova et al., 2017*; *Mittler, 2006*; *Razin et al., 2012*). So far, a total of 176, 189, 124, 109, 118, and 321 C2H2-ZFPs have been identified in *Arabidopsis thaliana* (*Englbrecht, Schoof & Bohm, 2004*), rice (*Oryza sativa*) (*Pinky et al., 2007*), foxtail millet (*Setaria italica*) (*Muthamilarasan et al., 2014*), poplar (*Populus trichocarpa*) (*Liu et al., 2015*), tobacco (*Nicotiana tabacum*) (*Yang et al., 2016*), and soybean (*Glycine max*) (*Yuan et al., 2018*), respectively. Moreover, two main structural features were widely detected in the C2H2-ZFPs of most plants. In comparison with those of yeast and animals, the plant C2H2-ZFP zinc-finger domains are commonly separated by a longer and more variable spacer between the two zinc fingers. In addition, the highly conserved "QALGGH" sequence is also unique to plant C2H2-ZFPs (*Cao et al., 2016*; *Ding et al., 2016*; *Wang et al., 2018*; *Zhang et al., 2016*). Subsequently, different types of C2H2-ZFPs have been defined in plants, including rice, *Arabidopsis*, petunia (*Petunia hybrida var. Mitchell diploid*), poplar, and soybean (*Agarwal et al., 2007*; *Englbrecht, Schoof & Bohm, 2004*; *Kubo et al., 1998*; *Liu et al., 2015*; *Yuan et al., 2018*); however, few C2H2-ZFs have been molecularly characterized in tomato (*Solanum lycopersicum*) (*Chang et al., 2018*; *Zhang et al., 2011*).

The first plant-specific C2H2 protein (ZPT2-1, renamed from EPF1) was identified in Petunia, and was found to interact with promoter region of the gene encoding 5-enolpyruvylshi-kimate-3-phosphate synthase (*Takatsuji et al., 1992*). Since then, the structures and functions of the plant C2H2-ZFPs have been widely reported, and shown to be involved in a variety of processes, including plant growth and development and the response to stresses (*Huang et al., 2009*; *Iida et al., 2000*; *Kim et al., 2016*; *Sun et al., 2010*; *Wang et al., 2016*). In *Arabidopsis*, the overexpression of zinc finger of *Arabidopsis thaliana* 12 (*ZAT12*) improved osmotic stress tolerance, and also interacted with ZAT7 or ZAT10 to enhance tolerance to salinity (*Li et al., 2018*; *Mittler et al., 2006*; *Sholpan et al., 2005*; *Sultan et al., 2007*). The *AZF1* (*Arabidopsis* zinc-finger protein 1), *AZF3*, and *STZ* (salt tolerance zinc finger genes) are associated with the cold-stress response in *Arabidopsis* (*Sakamoto et al., 2000*), while *ZAT18* was found to positively modulate drought-stress tolerance (*Yin et al., 2017*). In rice, the C2H2-ZFPs play a role in many aspects of stress tolerance, regulating the responses to cold, drought, oxidative, and salt stresses (*Huang et al., 2009*; *Sun et al., 2010*; *Xu et al., 2008*; *Zhang et al., 2014*). The functions of the C2H2-ZFPs in many other plants have also been reported, and are often found to be involved in plant development processes such as morphogenesis of cell and trichomes, ripening and senescence (*Chang et al., 2018*; *Moreau et al., 2018*; *Weng et al., 2015*), and stress resistance such as aluminum, drought and salt (*Atreyee et al., 2018*; *Li et al., 2018*; *Rai, Singh & Shah, 2012*). These findings showed that the C2H2-ZFPs are active in multiple physiological

processes under stress conditions; however, the functions of the majority of the C2H2-ZFPs in tomato have not been reported.

Tomato is considered a model system for the study of both fleshy fruit development and the biology of the Solanaceae species (*The Tomato Genome, 2012*), and is resistant to a wide range of abiotic stress conditions (*Yáñez et al., 2009*). Previous studies have revealed that the C2H2-ZFPs play important roles in the defense and acclimation responses of plants to various environmental stress conditions (*Wang et al., 2018*); therefore, it is important to complete the genome-wide identification and expression analysis of the tomato *C2H2-ZF* family to better understand their roles in stress responses. Here, we identify 104 *C2H2-ZFs* in tomato and provide a comprehensive analysis of their phylogenetic relationships, genomic locations, and gene structures. Furthermore, the expression profiles of this gene family were analyzed in different tissues and under high temperature stress using data from heatmaps and quantitative real-time reverse transcription polymerase chain reaction (qRT-PCR) analyses. This enabled us to reveal the transcriptional regulatory mechanisms of the *C2H2-ZFs* in tomato, and will provide valuable information for future cloning and functional studies of these genes in tomato and other Solanaceae species.

## MATERIALS & METHODS

### Genome-wide analysis of the C2H2-ZF family in tomato

The tomato C2H2-ZFP family members were identified using their homology to the *Arabidopsis thaliana* C2H2-ZFP sequences from the TAIR10 database (File S1). The tomato genome (*The Tomato Genome, 2012*) and protein sequences (https://solgenomics.net/organism/Solanum_lycopersicum/genome) were used to construct a local protein database using Geneious v4.8.5 software (http://www.geneious.com/; Biomatters, Auckland, New Zealand), which was then BLASTP searched using the sequences of the *A. thaliana* C2H2-ZFPs (obtained from the Arabidopsis Information Resource; TAIR10, https://www.arabidopsis.org) as queries, with an $E$-value cut-off $\leq 1 \times 10^{-20}$. Subsequently, the Hidden Markov Model (HMM) profiles of the C2H2-ZFP sequences (Pfam ID: PF00096) were downloaded from the Pfam database (http://pfam.xfam.org) and used to validate the identity of all candidate *C2H2-ZF* gene members.

All the obtained C2H2-ZFP sequences were further confirmed using the NCBI Conserved Domain Database (CDD; https://www.ncbi.nlm.nih.gov/cdd/) with the default parameters. Proteins without C2H2-ZF domains were removed. The locus ID and chromosomal location information of each tomato *C2H2-ZF* gene family member were obtained from the genome annotation file (File S1), and the lengths of the coding sequences (CDSs) were determined by performing BLASTn searches against the Sol Genomics Network database (https://solgenomics.net). The physicochemical properties of the deduced proteins, including the molecular weight (MW), isoelectric point (pI), and grand average of hydropathy (GRAVY) values, were determined using the ExPASy-ProtParam tool (http://web.expasy.org/protparam/).
## Phylogenetic analysis and gene duplication

To identify the evolutionary relationships of the *C2H2-ZF* gene family members, all C2H2-ZFP sequences were aligned using ClustalW2 software under the default settings (*Larkin et al., 2007*). The aligned sequences were then subjected to a phylogenetic analysis using MEGA v6.0 (Tokyo Metropolitan University, Tokyo, Japan; *Tamura et al., 2013*). Subsequently, we compared the phylogenetic trees constructed and tested by different methods, including Maximum Likelihood (ML), Neighbor Joining (NJ), unweighted pair-group method with arithmetic means (UPGMA) and Maximum Parsimony (MP) methods, respectively. The phylogenetic trees were constructed using different methods with 1,000 replicate bootstrap tests. The ML trees was calculated using the ProtML program under the JTT model (*Guo et al., 2008*), NJ trees was obtained using the JTT+I+G substitution model (*Pan et al., 2018*), UPGMA and MP trees were generated in MEGA v6.0 (*Tamura et al., 2013*) with the default parameters. Finally, the phylogenetic trees were visualized using FigTree v1.4.2 (http://tree.bio.ed.ac.uk/software/figtree/).

Based on the GFF genome files from the Sol Genomics Network database (https://solgenomics.net; *The Tomato Genome, 2012*), a map of the distribution of the C2H2-ZF genes was drawn for each chromosome using MapChart v2.0 (*Voorrips, 2002*).

## Gene structure, conserved motif and functional annotation analyses

Genome DNA and the corresponding CDSs of the putative *C2H2-ZF* genes were obtained from the Sol Genomics Network database (https://solgenomics.net; *The Tomato Genome, 2012*), then analyzed using the Gene Structure Display Server (GSDS v2.0; http://gsds.cbi.pku.edu.cn/index.php) to obtain information on the exon/intron structures. MEME v5.0.3 (http://meme-suite.org/tools/meme) was used to predict the corresponding conserved motifs (*Bailey et al., 2009*) with the following parameters: optimum motif widths of 6–300 residues, any repetition, and a maximum of 10 motifs (*Pan et al., 2018*). Each motif with an $E$-value $<1 \times 10^{-10}$ was retained for motif detection. In addition, gene ontology analysis of tomato *C2H2* family genes was performed using the Blast2GO program (*Conesa et al., 2005*) with the default parameters.

## Expression analysis of the C2H2-ZF family members

The expression profiles of the *C2H2-ZF* genes were measured using the publicly available tomato RNA-Seq datasets retrieved from the TomExpress database v17.0.0 (Available online: http://tomexpress.toulouse.inra.fr/; *Zouine et al., 2017*). Subsequently, the expression levels had been normalized in tomato as the published method (*Maza et al., 2013*), and the pipeline was described below. The reads were mapped to the tomato genomes SL2.40 by the TopHat 2.0.0 software with default parameters (*The Tomato Genome, 2012*; *Trapnell et al., 2012*). Gene expression levels were assessed with Cufflinks software with default parameters and using the tomato gene annotation file ITAG3.2 (*The Tomato Genome, 2012*; *Trapnell et al., 2012*). Eventually, the visualized heatmaps were generated using Heatmap Illustrator 1.0.

## Plant materials and heat treatment

The wild-type tomato (*Solanum lycopersicum* cv. Ailsa Craig) was grown in 15-cm pots containing a soil mix of humus, vermiculite, and perlite in a ratio of 3:1:1. The plants were grown under standard glasshouse conditions (16 h light and 8 h dark at 25 $\pm$ 2 °C). After one month of cultivation, the plants were subjected to a heat stress condition (42 °C) for 12 h or 24 h, while those collected at 0 h were used as the controls. All samples (leaves, stems, and roots) were collected and immediately frozen in liquid nitrogen for RNA extraction. Three biological repetitions were performed for each treatment.

## RNA isolation and qRT-PCR analysis

Total RNA was isolated from the roots, stems, and leaves using the RNAiso Plus Kit (TaKaRa Bio, Dalian, China). 1-μg aliquot of RNA was treated with DNase I (Takara Bio, Dalian, China) and transcribed into cDNA using the Oligo dT-Adaptor Primers and an RNA PCR Kit (AMV) v3.0 (Takara Bio, Dalian, China). The qRT-PCR analysis was performed on a CFX96 Real-Time PCR system (Bio-Rad Laboratories, Hercules, CA, USA) using Eva Green SMA (Bio-Rad Laboratories). To evaluate relative gene expression levels, the *SlEF1-α* gene *(Solyc06g005060)*, *SlACT (Solyc03g078400)*, and *SlUBI3 (Solyc01g056940)* were used as the internal reference, and 35 pairs of gene-specific primers were used for the qRT-PCR (Table S1). The relative expression levels of these genes were calculated using the $2^{-\Delta\Delta}$ method (*Pan et al., 2012*). To ensure the statistical credibility of the results, all experiments were performed with three biological replicates and three technical replicates. And data were compared using $t$-test.

# RESULTS

## Genome-wide identification and characterization of the *C2H2-ZFs* in tomato

Using BLAST and the HMM profiles of the C2H2-ZF domain, a total of 116 tomato cDNAs in the tomato genome (cDNA release 3.20; https://solgenomics.net/organism/Solanum_lycopersicum/genome) were annotated as encoding C2H2-ZFPs. All the C2H2-ZF candidates were analyzed using the NCBI Conserved Domain Database to verify the presence of the C2H2-ZF domain. Finally, 104 C2H2-ZFPs were confirmed in tomato (File S1). Correspondingly, their physical and molecular properties, including their Soly IDs, chromosome information, genomic positions, lengths of both the CDSs and proteins, theoretical isoelectric points, and molecular weights were predicted (Table S2). All identified C2H2-ZF genes were found to encode proteins varying from 96 to 1,178 amino acids, including a few exceptionally longer (*Solyc04g074250.2.1*) or smaller proteins (*Solyc00g015730.2.1*). Correspondingly, their molecular weights varied from 11.021 kDa (*Solyc00g015730.2.1*) to 128.573 kDa (*Solyc04g074250.2.1*), and their predicted isoelectric points (pI) ranged from 4.49 (*Solyc10g085560.2.1*) to 10.62 (*Solyc00g014800.1.1*). Detailed information of tomato *C2H2-ZF* family is shown in Table S2.

## Comparative phylogenetic analysis of the *C2H2-ZFs*

To explore the phylogenetic relationships of the C2H2-ZFPs between tomato and the model plant *Arabidopsis*, a NJ phylogenetic tree was firstly constructed from an alignment

of the 104 tomato C2H2-ZFPs and 97 *Arabidopsis* C2H2-ZFPs (Table S3). The sequences were clustered into six major groups (Fig. 1A, Group 1-6, Table S3), each of which could be further subdivided into different subgroups according to their bootstrap values. Furthermore, the tomato C2H2-ZFPs were tightly grouped with the *Arabidopsis* C2H2-ZFPs (Groups 1 to 4, and Group 6; Fig. 1A), except for the Group 5 (Fig. 1A), indicating that we could assume the putative function of these genes in tomato based on the Arabidopsis C2H2-ZFPs in same group. Subsequently, we reconstructed the phylogenetic distribution of the 104 *C2H2-ZF* genes using different methods to clearly elucidate the relationships between the *C2H2-ZF* family genes in tomato. Results showed that the phylogenetic tree constructed by different methods displayed the differences, such as NJ tree (Fig. 1B), ML tree (Fig. S1A), UPGMA tree (Fig. S1B) and MP tree (Fig. S1C). For the NJ phylogenetic tree, 104 C2H2-ZFPs were classified into nine groups (I to IX; Fig. 1B). Among them, four genes in Group VII (Fig. 1B) were all grouped in Group 1 (Fig. 1A), some genes in Group VII, VIII and IX (Fig. 1B) was closely related to Group 1 and 2 (Fig. 1A), and Group II, III, IV (Fig. 1B) was closely related to Group 3 (Fig. 1A), while a small number of C2H2-ZFPs were clustered into the different groups in tomato (Figs. 1A and 1B). These results suggested that the same clades of phylogenetic tree were likely to represent the closest homologous gene pairs between tomato and Arabidopsis, and *C2H2-ZFs* have also undergone sequence diversification independently in different organisms. In addition, 104 *C2H2-ZFs* were irregularly distributed between these groups in tomato (Fig. 1B), such as Groups II, VIII, and IX were the main clades, containing 18, 26, and 28 genes, respectively, while Groups I, IV, and VII were the smallest clades, containing three, three, and four tomato *C2H2-ZFs*, respectively (Fig. 1B).

## Conserved domain analysis of the C2H2-ZFPs

The ZF domain structure is X2-Cys-X(2-4)-Cys-X12-His-X(3-5)-His, where X represents any amino acid and the numbers indicate the consensus spacing between the conserved amino acid residues. This is highly conserved and essential for the ZF configuration and loop stability. To investigate the characteristics of the C2H2-ZF domains in tomato, the 104 C2H2-ZFPs were analyzed using the NCBI Conserved Domain Database (CDD; https://www.ncbi.nlm.nih.gov/cdd/). Multiple protein sequence alignments revealed that the ZF domains in all of these proteins were highly conserved. Among them, 83 C2H2-ZFPs (about 79.81%) contained a X2-Cys-X2-Cys-X12-His-X3-His sequence, seven C2H2-ZFPs (about 6.73%) had a X2-Cys-X2-Cys-X12-His-X4-His sequence, and the remains showed different types of C2H2-ZF domains (Fig. 2). In addition, 50 C2H2-ZFPs with a plant-specific conserved domain 'QALGGH' were identified in tomato and mainly classified into Groups VIII and IX (Fig. 2), which were fewer than that in Arobidopsis (64) (*Englbrecht, Schoof & Bohm, 2004*), rice (65) (*Agarwal et al., 2007*), poplar (62) (*Liu et al., 2015*) and foxtail millet (97) (*Muthamilarasan et al., 2014*), respectively.

## Analysis of the exon–intron structures and conserved motifs in the tomato *C2H2-ZFs*

The Gene structural diversity within a gene family can be used as an evolutionary marker (*Zhu et al., 2018*). To gain further insights into the structural diversity of tomato *C2H2-ZFs*,

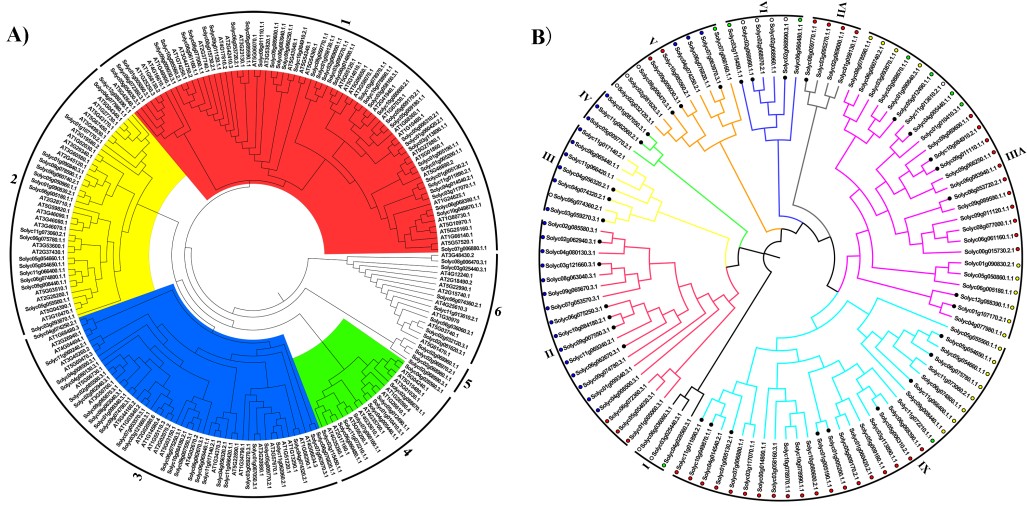

**Figure 1 The evolutionary relationship of the *C2H2-ZFs*.** The neighbor-joining tree was created using the MEGA6.0 program (bootstrap value set at 1,000). (A) The phylogenetic tree representing the relationships among 201 C2H2-ZFPs of tomato and Arabidopsis. All C2H2-ZFP sequences were grouped into six groups (A to F), which are represented by different colors. (B) The phylogenetic tree representing relationships among 104 C2H2-ZFPs of tomato. All C2H2-ZFP sequences were grouped into nine groups (I to IX), which are represented by different colors. The different color dots in Fig. 1B were identical with the same clades in Fig. 1A. The *C2H2-ZFs* used in the expression analysis following the heat stress treatment are marked with solid black circles. The evolutionary relationship of the *C2H2-ZFPs*.

we compared the exon–intron structure of each of the *C2H2-ZFs*. The number of introns in the *C2H2-ZFs* varied from 0 to 10. Based on the number of introns, 65 *C2H2-ZFs* (62.5%) was intronless, 29 *C2H2-ZFs* had 1 to 3 introns (27.9%), and the remaining *C2H2-ZFs* contained more than four introns (9.6%). In general, genes in the same group shared similar intron/exon arrangements in terms of intron numbers and exon length. For example, most *C2H2-ZFs* in Groups III, IV, VI, VII, VIII and IX genes had zero to one intron, and in Group II had two to three introns, which had the normal intron length (Fig. 3, Table S2). In contrast, the gene structure appeared to be more variable in groups I and V, which had striking distinctions in the exon/intron structure variants (Fig. 3, Table S2). In addition, we compared the exon–intron structure of each of the *C2H2-ZFs* in the same cluster of the different phylogenetic tree constructed by NJ (Fig. 3), ML (Fig. S1A), UPGMA (Fig. S1B) and MP (Fig. S1C), respectively. We found that genes in the same group had the highly similar intron/exon arrangements in the NJ phylogenetic tree (Fig. 3), indicating that the gene structure patterns were consistent with the NJ phylogenetic analysis.

Using the MEME tool, a total of 10 conserved motifs were identified in the C2H2-ZFPs, and the lengths of these conserved motifs varied from 7 to 31 amino acids (Fig. 4, Fig. S2). Among them, Motif 1 was widely detected in all C2H2-ZFPs, corresponding to the C-X2-C-X12-H-X3-H single ZF structure, which located at the N-terminal region of C2H2-ZFPs in groups II, and VI-IX (Fig. 4). Some groups also contained specific motifs; for example, Motifs 2 and 3 were present in the N-terminal region of Group II and the C-terminal region of many members in Groups III and IV; while Motif 4 was only detected

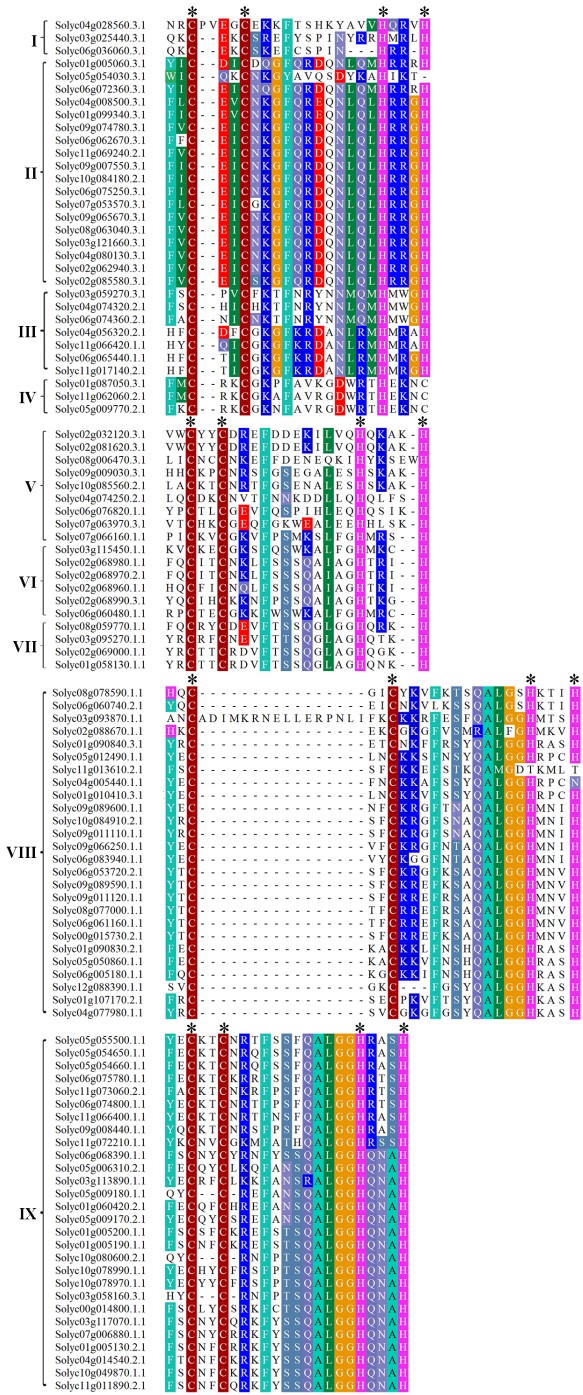

**Figure 2   Multiple protein sequence alignments of the C2H2-ZF domains in the tomato C2H2-ZFPs.**
The proteins were categorized into nine groups (I to IX) based on NJ phylogenetic tree, as shown on the left of the C2H2-ZFPs. The identical and conserved amino acid residues were represented by colored backgrounds, respectively. The position of conserved C2H2 domain was represented by the asterisk.

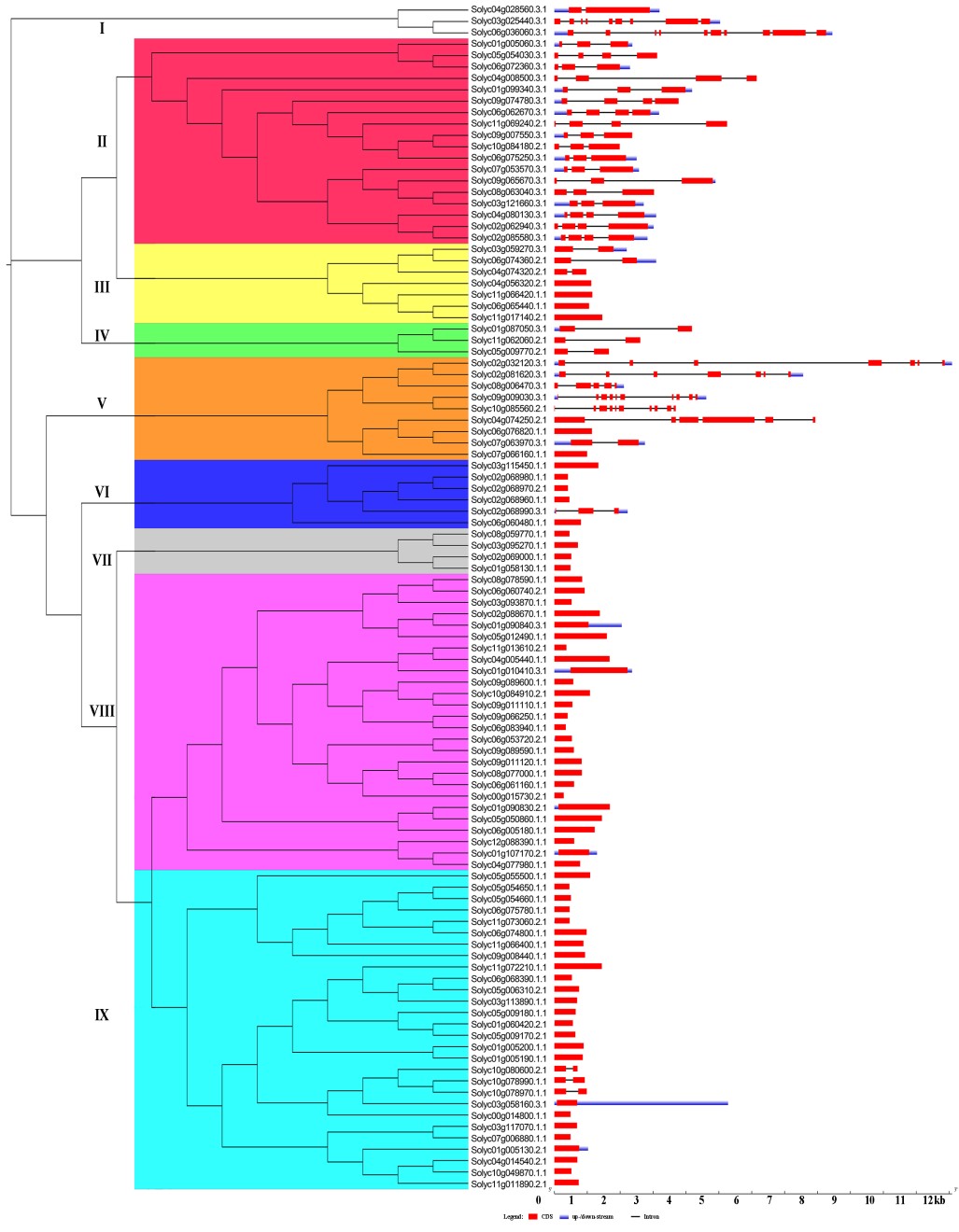

**Figure 3 Gene structure of the tomato *C2H2-ZF* family members.** The coding sequences (CDSs), untranslated regions (UTRs), and introns are depicted by filled red boxes, blue boxes, and single black lines, respectively. Groups (I to IX) are indicated on the left of the *C2H2-ZF*s. The scale bar indicates the length of the corresponding genes (kb).

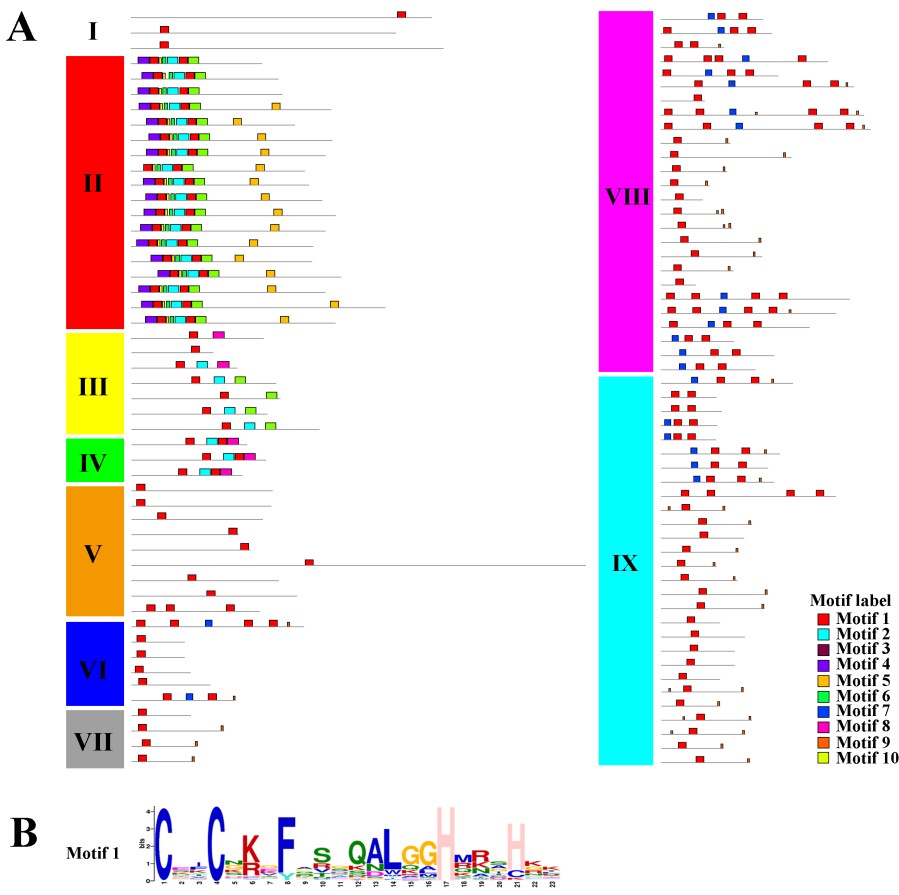

**Figure 4** **The conserved and potential motifs in the tomato C2H2-ZFPs.** (A) The distribution of 10 conserved motifs in the C2H2-ZFPs, are identified using MEME v5.0.3 and displayed in different colored boxes. (B) The sequence of Motif 1 corresponds to the C-X2-C-X12-H-X3-H single ZF structure. The detailed motif sequences are shown in Fig. S2.

in the N-terminal region of Group II, indicating that they may be relevant to the specific functions of these genes. The members of Groups I and V–IX have relatively simple motif patterns in comparison with Groups II to IV, implying the possible functional divergence of the *C2H2-ZF* genes in tomato.

Taken together, the structure and motif conservation within the *C2H2-ZF* genes supports the results of the NJ phylogenetic analysis. Variations in the motif compositions between subfamilies might be explained by their functional diversification.

## Chromosomal locations and gene duplication events in the *C2H2-ZFs*

To gain insight into the organization of these genes in the tomato genome, the 104 *C2H2-ZFs* were mapped onto their respective chromosomes, which acquired from tomato genome database (*The Tomato Genome, 2012*). The *C2H2-ZF* genes were unevenly distributed throughout the tomato genome and generally more abundant at the both ends of the chromosomes (Fig. 5, Table S2). Chromosome 6 contained the largest number of *C2H2-ZFs* (16), followed by chromosomes 1, 2, 3, 4, 5, 9, 10, and 11, each of which contained 7

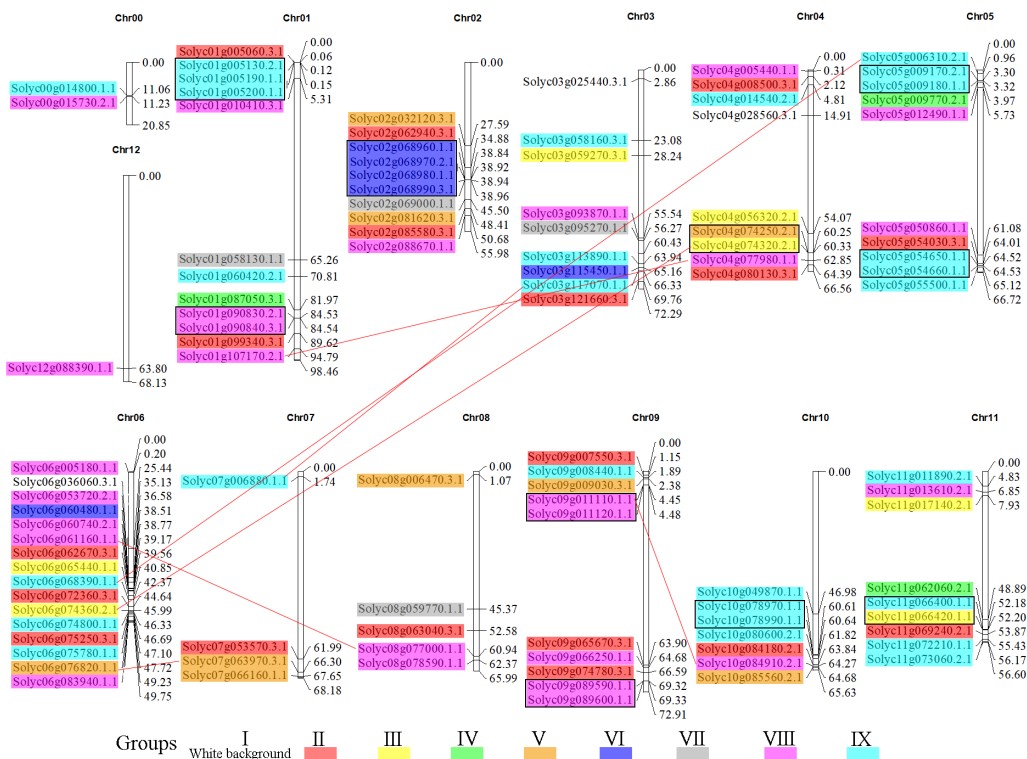

**Figure 5  Genomic distribution of the *C2H2-ZFs* across the tomato chromosomes.** The chromosome numbers and sizes (Mb) are indicated at the top and right of each bar, respectively. Genes from the same subgroups are indicated by the same color, which is consistent with the coloration used on the phylogenetic tree (Fig. 1B). The *C2H2-ZF* gene pairs resulting from segmental duplication genes are connected by red lines, and the tandem duplication gene clusters are marked in black rectangles.

to 12 *C2H2-ZF* s. Chromosomes 0 (random chromosome) and 12 contained two and one genes, respectively. In addition, the duplication events of *C2H2-ZFs* were also analyzed in tomato genome since gene replication play an important role in genomic expansions and realignments. We identified 15 pairs of tandem-duplicated gene pairs (with two or more homologous genes within 100 kb region) located on chromosomes 1, 2, 4, 5, 9, 10, and 11 (Fig. 5), respectively. The closely related clustered sequences of Groups VI, VIII, and IX are mainly located on chromosomes 1, 2, 5, 9, 10, and 11, suggesting that the expansion of this gene family may have occurred via localized or intra-chromosomal duplication. In addition to tandem duplications, seven segmental duplication events were detected to scatter in eight chromosomes (Fig. 5). Furthermore, many homologous genes were located in different chromosomes in tomato, supporting the high conservation of the *C2H2-ZF* gene family.

## Functional annotation of *C2H2-ZFs*
All 104 *C2H2-ZFs* were subjected to GO enrichment for investigating their functional annotation. As a result, a total of 43 *C2H2-ZFs* were mapped on the GO database, resulting in 285 annotations (Table S5), which were distributed across three ontology categories

(biological processes, cellular components, and molecular functions) with 30 function terms (Fig. S5). Furthermore, most tomato *C2H2-ZFs* were annotated as being associated with meal ion binding, DNA-binding transcription factor activity, nucleus and regulation of transcription, DNA-templated, respectively (Fig. S5).

## Expression patterns of the *C2H2-ZFs* in various tissues and organs

Using the TomExpress database (available online: http://tomexpress.toulouse.inra.fr/), we investigated the expression levels of the 104 *C2H2-ZFs* in various tissues of the tomato cultivar Micro Tom, including its roots, leaves, flower buds, petals, flowers, fruits, flesh, and peel, which were visualized using a heatmap (Fig. 6). The *C2H2-ZFs* had different expression patterns across the various tissues. Firstly, the clear differences in expression between the genes of Group II, III and V in comparison with those of the other groups were consistent with their different gene structures and the phylogenetic tree analysis (Figs. 3 and 6). In addition, we found that a small few genes typically had relatively high expression levels across various tissues and organs at different development stages, such as *Solyc01g099340.3.1* in Group II, *Solyc09g009030.3.1* in Group V, *Solyc04g077980.1.1* in Group VIII, and *Solyc06g075780.1.1* in Group IX, while most genes of Groups IV, VI, VII, and VIII had relatively low expression levels (Fig. 6). Some genes, including *Solyc05g054030.3.1*, *Solyc05g055500.1.1*, *Solyc07g063970.3.1*, *Solyc09g011110.1.1*, *Solyc10g080600.2.1*, and *Solyc11g066400.1.1*, were specially expressed in the different tissues, and genes (*Solyc01g005130.2.1*, *Solyc04g014540.2.1*, *Solyc06g075780.1.1*, and *Solyc07g006880.1.1*) in Group IX showed similar expression patterns during the fruit development (Fig. 6). However, the expression patterns of a few gene pairs, including *Solyc03g025440.3.1*, *Solyc04g056320.2.1*, *Solyc06g065440.1.1*, and *Solyc11g017140.2.1*, were significantly different in Groups I and III, although they were paralogous genes (Figs. 1B and 6).

## Expression of the *C2H2-ZFs* under heat stress

To improve the reproducibility and reliability of the qRT-PCR analysis obtained in this study, three previously reported tomato internal reference genes, including *SlEF1-α (Solyc06g005060)*, *SlACT* (*Solyc03g078400*), and *SlUBI3* (*Solyc01g056940*), were primarily detected for the their quantification stability in the leaves, stems, and roots under the control and heat stress by qRT-PCR method. The results showed that *SlEF1-α* (Fig. S3A) displayed relatively greater stability than *SlACT* and *SlUBI3* among the different tomato tissues (Figs. S3B and S3C, Table S4), which were used for further analysis in this study. Many of the *C2H2-ZFs* previously described in other crops were involved in responding to abiotic stresses, such as drought, heat, and salt stress (*Muthamilarasan et al., 2014*; *Sakamoto et al., 2004*; *Sun et al., 2010*; *Wang et al., 2018*). To confirm the potential roles of candidate tomato *C2H2-ZFs* in heat stress, the expression profiles of 34 *C2H2-ZFs* were finally detected in leaves, stems and roots of tomato under heat stress (42 °C), which might not be markedly induced by the day/night cycle, except the *Solyc10g085560.2.1* (Fig. S4). Combined with the expression phylogenetic analysis, we found that genes showed the diversity expression patterns under heat stress (Figs. 7 and 8). Of them, the expression

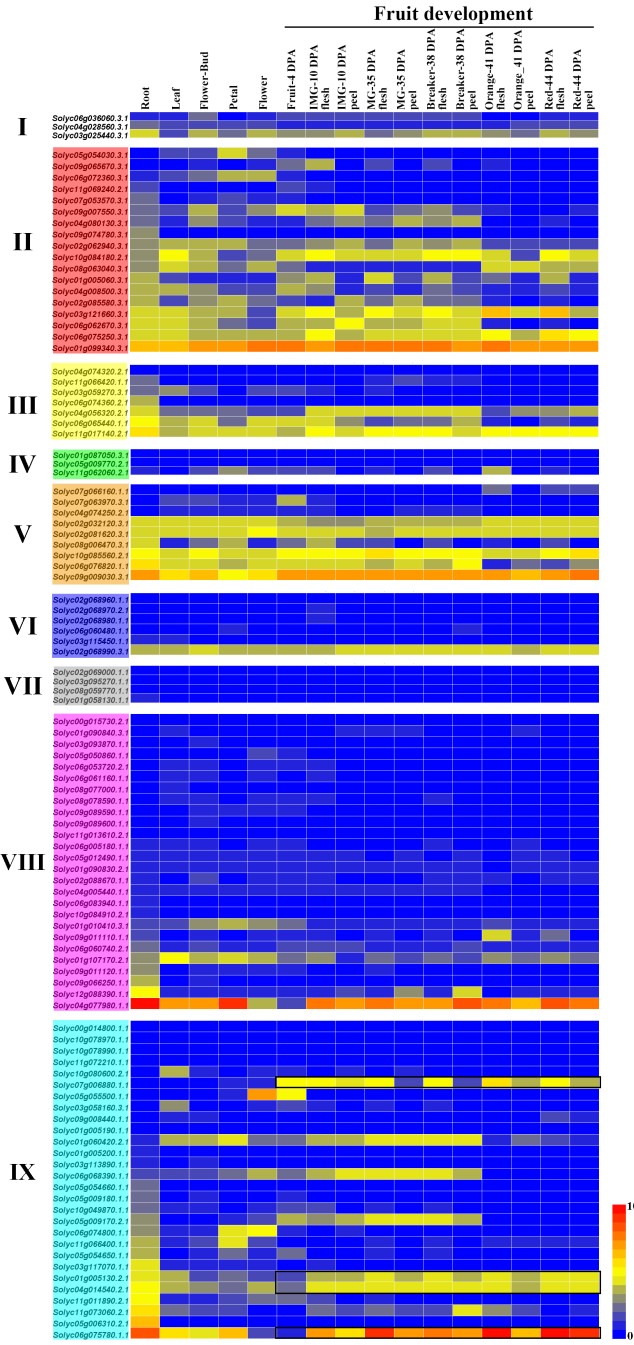

**Figure 6  Relative expression of the *C2H2-ZF*s in various tissues and growth stages of tomato.**  The expression profiles were generated from transcriptomic data v17.0.0 (Available online: http://tomexpress.toulouse.inra.fr/; *Zouine et al., 2017*). Genes from the same subgroups are indicated by the same color, which is consistent with the I-IX used on the phylogenetic tree (Fig. 1B). The textbox indicates the *C2H2-ZF* genes showed the similar expression patterns during the fruit development from Group IX. The color scale (0 to 10, dark blue to red) represents the standardized gene expression levels.

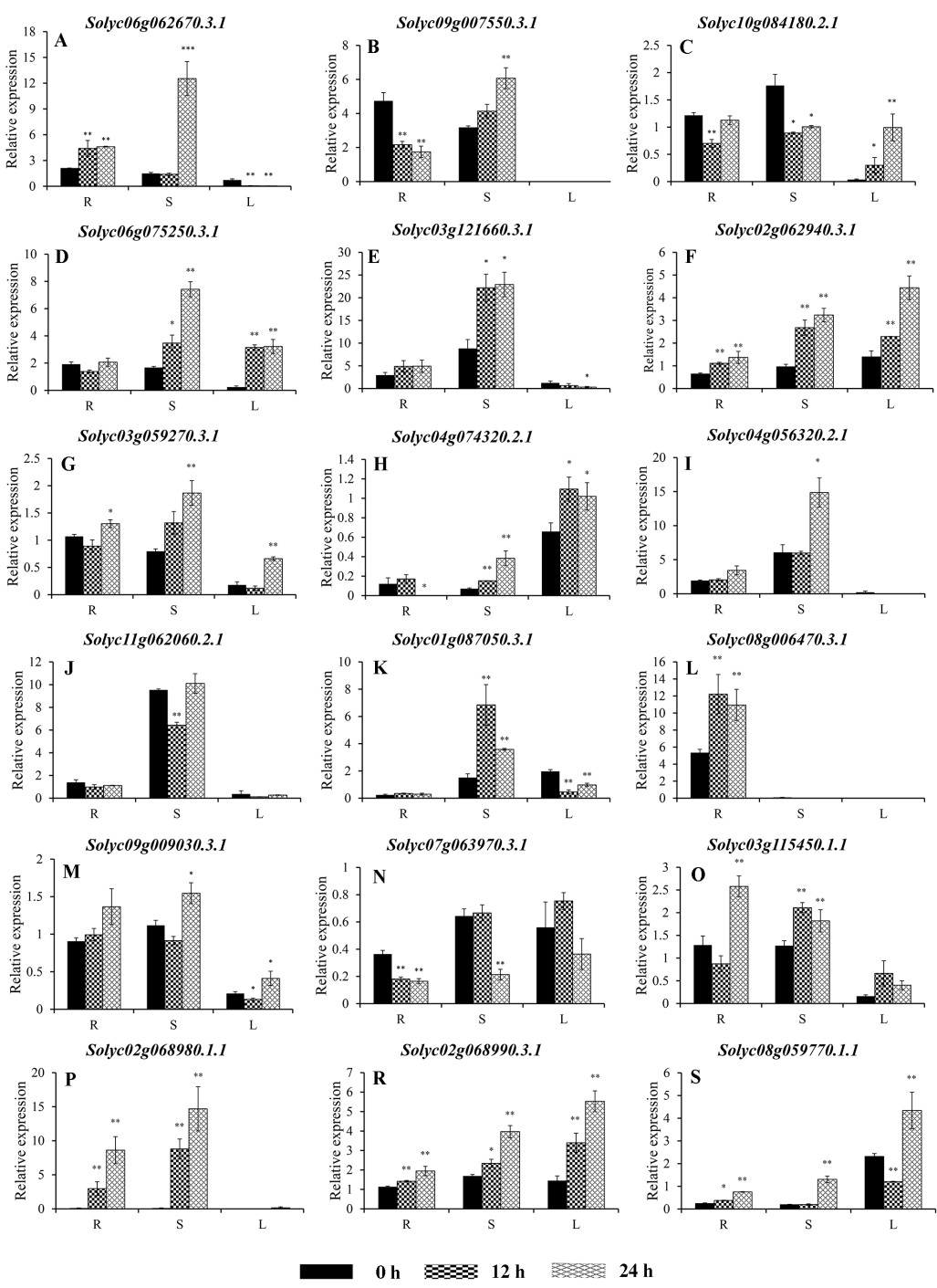

**Figure 7** **Expression analysis of 18 selected *C2H2-ZFs* from Group I to VII in the roots, stems, and leaves of tomato plants under heat stress, revealed using qRT-PCR.** Values represent the average ± SD of three biological replicates with three technical replicates of each reaction. Error bars represent the standard deviations from three biological replicates. The relative expression levels were normalized according to the reference gene (*SlEF1-α*; *Solyc06g005060*) to the values in control (0 h). Data were compared using Student's *t*-test: *, $P < 0.05$ and **, $P < 0.01$, respectively. R, S, and L indicate the roots, stems, and leaves, respectively. A–F, G–I, J–K, L–N, O–R, and S represents genes were classified into Group II, III, IV, V, VI, and VII, respectively.

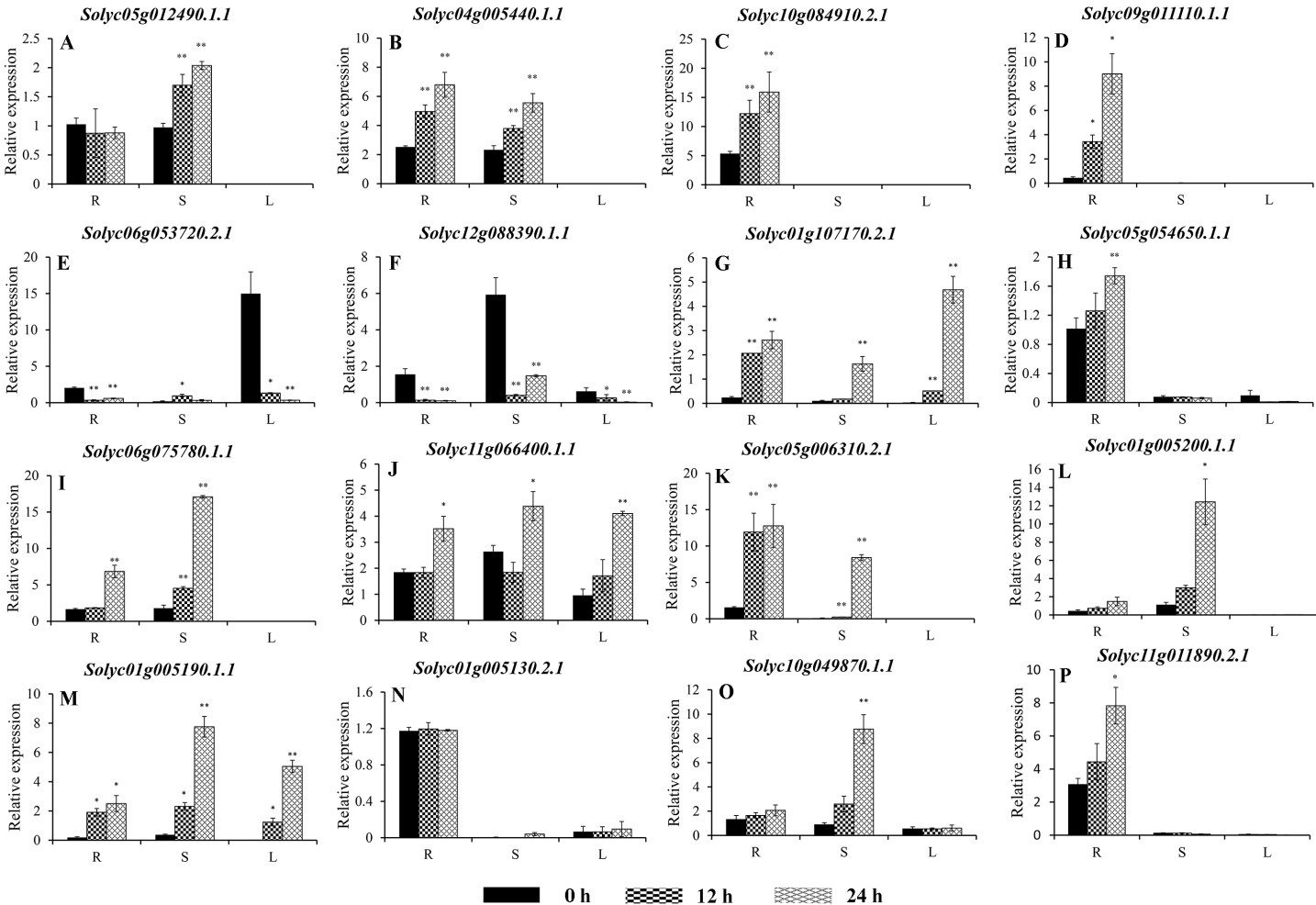

**Figure 8 Expression analysis of 16 selected *C2H2-ZFs* from Group VIII and IX in the roots, stems, and leaves of tomato plants under heat stress, revealed using qRT-PCR.** Values represent the average ± SD of three biological replicates with three technical replicates of each reaction. Error bars represent the standard deviations from three biological replicates. The relative expression levels were normalized according to the reference gene (*SlEF1-α*; *Solyc06g005060*) to the values in control (0 h). Data were compared using Student's *t*-test: *, $P < 0.05$ and **, $P < 0.01$, respectively. R, S, and L indicate the roots, stems, and leaves, respectively. A–G, and H–P represents genes were classified into Group VIII, and IX, respectively.

levels of 18 genes analyzed from Group II to VII were found to differ in roots, stems and leaves, under heat stress (Figs. 7A–7S). Of these, *Solyc02g062940.3.1* and *Solyc02g068990.3.1* were sustained up-regulated in the roots, stems, and leaves during heat stress treatment (Figs. 7F and 7R), and *Solyc02g068980.1.1* was significantly up-regulated in the roots and stems with the similar expression patterns (Fig. 7P), but *Solyc08g006470.3.1* was only significantly up-regulated by the heat treatment in the roots (Fig. 7L). For the largest classification groups VIII and IX, the expression levels of seven and nine genes from Group VIII and IX, were also investigated (Fig. 8). In Group VIII, *Solyc05g012490.1.1* and *Solyc04g005440.1.1* might be hard to induce by heat treatment with lower expression levels in leaves (Figs. 8A and 8B), while *Solyc10g084910.2.1* and *Solyc09g011110.1.1* showed the

similar expression patterns and only significantly up-regulated in the roots (Figs. 8C and 8D). In contrast, *Solyc06g053720.2.1* and *Solyc12g088390.1.1* were significantly suppressed in the roots, and leaves under heat treatment (Figs. 8E and 8F). Notably, the heat treatment induced significantly high levels of *Solyc01g107170.2.1* expression in all tissues (Fig. 8G). In Group IX, only *Solyc01g005190.1.1* was significantly up-regulated in all tissues after heat treatment (Fig. 8M), while *Solyc06g075780.1.1* and *Solyc05g006310.2.1* were strongly up-regulated in the roots and stems (Figs. 8I and 8K). Although these genes showed various expression patterns under heat treatment, most of them in the identical group had also similar expression patterns in the same tissues, such as Group II genes (*Solyc06g075250.3.1*, *Solyc03g121660.3.1*, and *Solyc02g062940.3.1*) in stems (Figs. 7D, 7E and 7F) and (*Solyc10g084180.2.1*, *Solyc06g075250.3.1*, and *Solyc02g062940.3.1*) in leaves (Figs. 7C, 7D and 7F), Group VII genes (*Solyc02g068980.1.1*, *Solyc02g068990.1.1*, and *Solyc08g059770.1.1*) in roots and leaves (Figs. 7P, 7R and 7S), and Group VIII genes (*Solyc05g012490.1.1* and *Solyc04g005440.1.1*) in the stems (Figs. 8A and 7B), which were significantly up-regulated under heat treatment, and Group VIII genes (*Solyc05g012490.1.1* and *Solyc04g005440.1.1*, *Solyc10g084910.2.1* and *Solyc09g011110.1.1*) in stems or leaves (Figs. 8A–8D) with lowly expression levels, while Group VIII genes (*Solyc06g053720.2.1* and *Solyc12g088390.1.1*) were significantly suppressed in roots and leaves (Figs. 8E and 8F), and so on.

Taken together, qRT-PCR was performed for 34 selected genes from different cluster groups under heat stress, and showed the various expression patterns in the roots, stems, or leaves. Among them, seven genes were strongly up-regulated in the roots, stems, and leaves, while one gene were down-regulated; and five genes up-regulated both in roots and stems, and five genes specially expressed in the roots, and so on (Figs. 7 and 8). Hence, these results suggest that these *C2H2-ZF* genes might be associated with the heat stress during the seedlings development in tomato.

## DISCUSSION

The C2H2-ZFPs are known to play important roles in many biological processes (*An et al., 2012*; *Joseph et al., 2014*; *Lyu & Cao, 2018*; *Wang et al., 2018*; *Weng et al., 2015*). To date, *C2H2-ZFs* have been identified and characterized in a variety of plant species, including soybean (*Yuan et al., 2018*), tobacco (*Yang et al., 2016*), *Arabidopsis* (*Englbrecht, Schoof & Bohm, 2004*), rice (*Agarwal et al., 2007*), poplar (*Liu et al., 2015*), and petunia (*Kubo et al., 1998*). Despite these advances, little was previously known about these genes in tomato, a model system for both fleshy fruit development and the Solanaceae species in general (*The Tomato Genome, 2012*). In this study, using 97 *Arabidopsis* C2H2-ZFP sequences as a query, we identified 104 *C2H2-ZF* family members in the tomato genome, which contains at least one C2H2-ZF motif (X2-Cys-X(2-4)-Cys-X12-His-X(3-5)-His), and the lengths of these sequences varied from 96 to 1,178 amino acid residues, with striking distinctions (Table S2), suggesting that a high degree of complexity among the tomato *C2H2-ZFs* (*Liu et al., 2015*). The 'QALGGH' motif was almost invariant in the tomato C2H2-ZFPs (Fig. 2), however, the C2H2-ZFP subfamilies in other plant species have previously been defined based on

changes to the conserved "QALGGH" motif (*Fedotova et al., 2017*; *Liu et al., 2015*; *Razin et al., 2012*; *Takatsuji, 1999*; *Wei, Si & Yang, 2016*; *Yuan et al., 2018*; *Zhang et al., 2016*). In this study, to reveal phylogenetic relationship of tomato C2H2 family members, 104 C2H2-ZFPs were divided into nine major groups according to the NJ phylogenetic tree (Fig. 1B), which were well consistent with arrangements, numbers, and types of their C2H2-ZF domains (Fig. 2). This suggests that the *C2H2-ZFs* were highly conserved during evolution and may have similar functions in tomato. However, classification of the *C2H2-ZFs* in NJ phylogenetic tree displayed the difference (Figs. 1A and 1B, Table S3), suggesting that they may show differences in gene function between the Arabidopsis and tomato.

The integration and rearrangement of gene fragments during evolution can lead to increases or decreases in the number of introns and exons present; therefore, the structural variation of genes is important for the evolution of gene families (*Xu et al., 2012*). Here, the gene structures and motifs present were highly similar among members of the same *C2H2-ZF* groups (Figs. 3 and 4); for example, the members of Groups I, II, and V had 3–5 exons (Fig. 3), while the most complex arrangements of motifs were observed in Group II (Fig. 4). These results were similar to those of previous analyses of the *C2H2-ZF* family genes in maize (*Zea mays*) (*Wei, Si & Yang, 2016*), soybean (*Yuan et al., 2018*), and poplar (*Liu et al., 2015*). Therefore, our results suggest that the sequences and biological functions of the C2H2-ZFPs were relatively conserved among members of the nine subgroups, indicating that our classification of the tomato C2H2-ZFPs was reasonable. About 62.5% had no introns and 27.9% had one to three introns with short length, which is consistent with the results that the genes with no intron or a short intron were tended to retain in plants (*Li & Liu, 2019*; *Mattick & Gagen, 2001*). Correspondingly, genes with fewer introns could be rapidly activated for respond to environmental challenges (*Jeffares, Penkett & Bähler, 2008*; *Li & Liu, 2019*), so *C2H2-ZFs* play an important roles in responding to abiotic stresses (*Muthamilarasan et al., 2014*; *Sakamoto et al., 2004*; *Sun et al., 2010*; *Wang et al., 2018*). In addition, previous results showed that C2H2-ZFPs with a plant-specific conserved domain 'QALGGH' play important roles in diverse environmental stress responses (*Agarwal et al., 2007*; *Kam et al., 2008*; *Liu et al., 2015*). In present study, about 48% (50 of 104) of C2H2-ZFPs from tomato had plant-specific conserved domain 'QALGGH', which was a larger number compared with other experimental models Arabidopsis (36%) and rice (34%), suggested that these C2H2-ZFPs are more important for tomato plants. Furthermore, most members in the same phylogenetic group had the similar intron/exon arrangements and motif compositions (Figs. 3 and 4). In addition, gene duplication events has been reported for *C2H2-ZF* gene families in different plants (*Agarwal et al., 2007*; *Guo et al., 2008*; *Yuan et al., 2018*), and which were also revealed the widely gene duplication events in the tomato genome, including 15 tandem duplication and 7 segmental duplication (Fig. 5). Thus, gene duplication events are one of the primary forces for the *C2H2-ZFs* gene evolution during the speciation and evolution of tomato. These results reflected the diverse functions of tomato *C2H2-ZFs* and will be helpful for their future functional analysis.

The tissue-specific expression of genes usually is preliminarily used to predict their corresponding functions (*Xiao et al., 2019*). Therefore, we assessed the expression profiles of the 104 *C2H2-ZFs* in various tomato tissues using published transcriptomic data

(*Zouine et al., 2017*), revealing that the *C2H2-ZF* genes display a diversity of relative expression patterns in different organs (Fig. 6) and may therefore play differing roles in various tissues or biological processes. But the homologous genes had similar expression patterns, such as *Solyc10g078970.1.1* (*SlHair*), *AT1G68360* (*GIS3*), *AT1G10480* (*ZFP5*), and *AT1G67030* (*ZFP6*) were all found to be grouped into Group 1 (Fig. 1A), which were previously reported to play important roles in controlling trichome development (*An et al., 2012*; *Chang et al., 2018*; *Sun et al., 2015*; *Zhou et al., 2013*). Reported genes, *AT5G04340* (*ZAT6*) and *AT1G27730* (*ZAT10/STZ*), *Solyc06g075780.1.1* (*SlZF3*) and *Solyc07g006880.1.1* (*SlZFP2*) were classified into the Group 2 (Fig. 1A) and Group IX (Fig. 1B), which showed the similar functions; for example, *AT5G04340* (*ZAT6*) and *AT1G27730* (*ZAT10/STZ*) was involved into the organs development and the adversity stress responses (*Devaiah, Nagarajan & Raghothama, 2007*; *Mittler et al., 2006*), and *Solyc06g075780.1.1* (*SlZF3*) enhanced the salt-stress tolerance in tomato (*Li et al., 2018*), and *Solyc07g006880.1.1* (*SlZFP2*) was characterized as a repressor to fine-tune ABA biosynthesis during fruit development (*Weng et al., 2015*), and they showed the higher expression levels during the fruit ripening in tomato (Fig. 6), which will be helpful for dissecting their roles in fruit ripening. In addition, the C2H2-Type Zinc Finger Protein, SUPPRESSOR OF FRIGIDA4 (SUF4, AT1G30970.3) could bind to the *Flowering Locus C (FLC)* promoter region, and play a role in transcriptional activation of *FLC* (*Kim et al., 2006*). In this study, the homologous genes, *Solyc02g032120.3.1* and *Solyc02g081620.3.1* belong to group V (Fig. 1B), and showed the relative high expression in the flowers (Fig. 6). These results suggested that the members within each group might have similar functions between tomato and Arabidopsis.

In nature, heat stress is one of the critical environmental factors that adversely affects plant growth and delays development (*Guan et al., 2013*; *Yang & Guo, 2014*). Several C2H2-ZPs were also characterized and participated in the interaction of plants and stress, such as *Zat6* (*Devaiah, Nagarajan & Raghothama, 2007*), *ZAT10/STZ* (*Mittler et al., 2006*), and C2H2 zinc-finger protein OsZFP213 (*Zhang et al., 2018*). To further characterize whether tomato *C2H2-ZF* genes play a role in heat-stress tolerance, the expression profiles of 34 *C2H2-ZF* genes randomly selected from all groups were analyzed in the roots, stems, and leaves of wild-type tomato under heat stress. As expected, the majority of the *C2H2-ZFs* were significantly up-regulated when the plants were exposed to heat treatment (42 °C), with different expression modes in the different tissues (Figs. 7 and 8). In plants, roots are known to play important roles in resisting abiotic stress, sensing soil changes and sending a series of signals to reduce root damage and maintain plant growth under abiotic stress (*Liu et al., 2015*; *Lynch, 1995*). The heat treatment used in the present study specifically induced the expression of 15 *C2H2-ZF* genes in the roots (Figs. 7 and 8); for example, *Solyc08g006470.3.1*, *Solyc10g084910.2.1*, and *Solyc09g011110.1.1* were strongly up-regulated in the roots at all time points, but not in the stem or leaves (Figs. 7L, 8C and 8D), indicating that they may play specific roles in the root responses to heat stress. Furthermore, *Solyc02g062940.3.1*, *Solyc02g068990.3.1*, and *Solyc01g005190.1.1* were significantly induced in all three tissues during all time points (Figs. 7F, 7R, and 8M), while *Solyc12g0883900.1.1* was significantly suppressed in the roots, stems, and leaves (Fig. 8F), suggesting these *C2H2-ZFs* might play important roles in mediating the response of tomato

plants to heat stress. In addition, previous study showed that the *Solyc06g075780.1.1* (*SlZF3*) could enhance salt-stress tolerance in tomato (*Li et al., 2018*), but we found that it was also induced by the heat treatment (Fig. 8I). These differences suggest that the tomato *C2H2-ZFs* may play a variety of roles in the response to heat stress, with the strongly suppressed genes potentially interacting synergistically with other genes involved in this process. Although the role of *C2H2-ZF* genes in these processes is not yet known, but the above-mentioned tomato *C2H2-ZF* genes with significant changes after heat treatment are useful in selecting candidate genes for functional validation in relation to heat stress in tomato.

## CONCLUSIONS

In this study, we characterized 104 *C2H2-ZFs* in the tomato genome using a genome-wide analysis. Examination of their phylogenetic relationships, chromosomal locations, gene structures, conserved motifs, and expression profiles revealed high levels of similarity between the identified subgroups of this family. This study lays the foundation for elucidating the functions of these important genes in future studies. In addition, the expression profiles of the *C2H2-ZFs* were evaluated during a heat stress treatment. These findings provide insight into the mechanisms of the *C2H2-ZF* function during the response to heat stress in tomato and potentially other Solanaceae species. Further molecular and functional analyses of these genes could suggest a strategy to improve heat tolerance to tomato plants.

## ACKNOWLEDGEMENTS

We would like to thank Kathy Farquharson for critical reading of this manuscript.

### Funding

This work was supported by the National Natural Science Foundation of China (NSFC) (No. 31772320), the Chongqing Natural Science Foundation (No. cstc2018jcyjA0947), the Chongqing Social Enterprise and People's Livelihood Guarantee Science and Technology Innovation Special Project (cstc2017shms-zdyfx0025), and Fundamental Research Funds for the Central Universities (No. XDJK2019C059 and XDJK2017D088). The funders had no role in study design, data collection and analysis, decision to publish, or preparation of the manuscript.

### Grant Disclosures

The following grant information was disclosed by the authors:
National Natural Science Foundation of China (NSFC): 31772320.
Chongqing Natural Science Foundation: cstc2018jcyjA0947.
Chongqing Social Enterprise and People's Livelihood Guarantee Science and Technology Innovation Special Project: cstc2017shms-zdyfx0025.
Fundamental Research Funds for the Central Universities: XDJK2019C059, XDJK2017D088.

## Competing Interests

The authors declare there are no competing interests.

## Author Contributions

- Xin Hu conceived and designed the experiments, performed the experiments, analyzed the data, prepared figures and/or tables, authored or reviewed drafts of the paper, approved the final draft.
- Lili Zhu performed the experiments, authored or reviewed drafts of the paper, approved the final draft.
- Yi Zhang analyzed the data, prepared figures and/or tables, approved the final draft.
- Li Xu performed the experiments, contributed reagents/materials/analysis tools, prepared figures and/or tables, approved the final draft.
- Na Li analyzed the data, contributed reagents/materials/analysis tools, prepared figures and/or tables, approved the final draft.
- Xingguo Zhang conceived and designed the experiments, authored or reviewed drafts of the paper, approved the final draft.
- Yu Pan conceived and designed the experiments, analyzed the data, prepared figures and/or tables, authored or reviewed drafts of the paper, approved the final draft.

## Data Availability

Raw data is available as a Supplemental File.

## Supplemental Information

Supplemental information for this article can be found online at http://dx.doi.org/10.7717/peerj.7929#supplemental-information.

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
