# Peer review of "Genome-wide identification of C2H2 zinc-finger genes and their expression patterns under heat stress in tomato (Solanum lycopersicum L.)"

_PeerJ, doi:10.7717/peerj.7929_

## Round 0.1 · original submission · Major Revisions

Based on the comments from the reviewers, your manuscript can be acceptable for publication after taking all comments from reviewers into consideration to make revision

Reviewer 1 ·

Basic reporting

1. My major concern about the number of identified putative members (104). The authors claimed that they have a great variation in their peptide length from 96 to 1,178 amino acid residues. Is all of them contain full C2H2 zinc figure motif, or some of them only partial sequence information? In figure 1, there might be some of the members are missing. A major group of figure 1 (VIII) do not possess C2H2 motif. What is the explanation for this? Moreover, figure 1 and 2 provide similar information. It will be better if the authors make figure S1 as figure 2, and explain the relation with Arabidopsis. However, the number of groups for tomato only is 9 (Figure 2), while this is only 6 in the case of tomato and Arabidopsis phylogenetic tree (Figure S1). Please explain these issues.

Experimental design

1. The authors stated that ‘We found that 65 C2H2-ZFs contained no introns (62.5%), 29 C2H2-ZFs had 1 to 3 introns (27.9%), and the remaining C2H2-ZFs contained more than four introns (9.6%). But there is no significance of studying presence or absence of intron. It is more interesting if the authors could explain the intron length variation among the genes due to evolution. In addition, kindly describe the analysis of tandem-duplicated gene pair’s identification.

Validity of the findings

1. What is the reason behind the separation of 34 gene expression profile data detected in leaves, stems, and roots of tomato under heat stress as figure 7 and 8?

Additional comments

2. Line 62, Put some references for the statement-
however, few C2H2-ZFs have been molecularly characterized in tomato (Solanum lycopersicum).

3. Kindly modify the writing style for the power value, such as
E-value cut-off 116 ≤ 1 ×10−20
E-value < 1 × 10−10

Reviewer 2 ·

Basic reporting

Xu et al., identified 104 C2H2-ZF genes in tomato based on their homology to Arabidopsis. They examined the genome distribution, phylogenetic relationships, gene structure, motif sequence conservation and expression patterns of these genes.Furthermore, they assayed the expression level of some C2H2-ZF genes under heat stress by qRT-PCR and found their association with heat stress. Generally speaking, this manuscript is well-written. The language, organization, intro & background and quality of figures are good. However, I have a few concerns in regards to the methods and interpretation of results. Since they don’t conflict with the main idea of this paper, I would suggest a minor revision.

Experimental design

The experimental design for this manuscript is generally fine. However, I have a major concern in regards to the expression analysis and heatmap. The authors need to provide details on how they normalized the expression levels derived from different studies. Usually normalization across different rna-seq experiments could be hard, thus I am not convinced about their heatmap. Additionally, they mentioned that their qRT-PCR results were broadly consistent with the RNA-seq data. They did not provide evidence explaining how they draw this conclusion. At least they should compare the RNA-seq data with the qPCR data side by side. They would need to specify how they convert the normalization of RNA-seq data for comparison and what was used as the control.

Validity of the findings

Statements that need to be clarified are listed as follows:
1.The authors did not elaborate how they classified 104 ZFPs into 9 clades in Figure 1. On Line 212, the authors stated that NJ method was consistent with the ‘expected results’. I guess this was referring to Figure 1. However, they should not claim Figure 1 is ‘expected’ as it was too subjective. They also need to explain what Figure S2A-C refers to here.
2. Line 217, the authors claimed Figure 2 was consistent with Figure S1. This point is not convincing. For example, there are 9 groups in Figure 2 whereas 6 groups in Figure S1.
3. Line 232, the authors did not provide information on how other methods is less consistent with Figure 3 compared to NJ. There is no grouping information provided for other methods. How did they perform the comparison?
4. L237, it would be a good idea if the authors would describe the position (N- or C- terminus) of ZF domain and the number of ZF domain in each group to support their grouping. Furthermore, it would be interesting to know the identity of the other motifs. ZF motif is often accompanied by other motifs, which could be important for their functions and classifications.

Additional comments

Minor points:
1.L194-121: It seems that the authors started with 104 genes and after removing proteins without ZF domains, there were still 104 genes. Please clarify on that.
2. Did not find the figure legend for supp figures.
3. Line 253, please elaborate how the tandem-duplicated gene pairs were identified.
4. L285-288. This sentence is confusing, please rephrase.
5. Figure 1, I would suggest the authors to highlight the C-C-H-H structure in the figure (e.g. use arrows).

Reviewer 3 ·

Basic reporting

The manuscript is written in scientifically appropriate English
The literature cited is ample and provides somewhat an adequate background on the paper’s topic. The figures are relevant, well labeled and well described. They are all high quality except for figure 3. The gene annotations are very hard to read as the letter font is too small. The authors should change the font and make it legible.
The structure of the paper is in the professional standard and appropriate. The objectives of the paper are clearly stated and discussed.

Experimental design

This manuscript aimed to identify C2H2-ZF genes from tomato and analyze their structure, available their phylogenetic relationships, genomic locations, and gene structures. Expression profiles of some of these genes under heat stress were also reported. The authors have taken advantage of the abundant amount of phylogenetic and expression data available in databases to identify 104 C2H2-ZF genes in tomato. The authors analyzed the structures of these genes and classified them into nine categories. The authors selected 34 of these 104 genes and performed expression analysis under heat stress. The authors then speculated on the functions of these genes based on their expression profiles.

Validity of the findings

I have issues with the way the authors interpret some of the expression data represented in figures 7 and 8. The authors should be very careful when talking about up and down regulation of genes. The writing results describing figures 7 and 8 in the results section is not completely accurate. That whole section should be carefully rewritten and I will be discussing the specifics in the comments below.

Additional comments

28 and 29: The sentence is incomplete. Please re-write
39: beta sheets instead of “beta sheet”
70. The sentence should read: and also interacted with ZAT7 or ZAT10 to enhance tolerance to salinity
80. Should read: the Stipules reduced (St) encodes a C2H2 zinc finger
91-92. Should read: the functions of the majority of the C2H2-ZFPs in tomato have not been reported (you have mentioned in the same paragraph, lines 82-86, that some of the C2H2-ZFPs have been already characterized)
285: The references in parenthesis have a different font than the rest of the paper.
286-288. Sentence starting at "To” in line 285 and ending at "(Figure S5) in line 288 is incomplete. Please re-write it
302 and 303: This is one example where the interpretation of the expression data is not accurate. The expression of Solyc11g062060.2.1 is not down regulated in the stems after 24 hours. Figure 7J shows that the levels of this gene are the same in 0 hours and 24 hours. It is down-regulated after 12 hours but the authors’ analysis is based on comparing the gene expression at 0 hours to 24 hours if I am correct.
307 and 308. The authors claim that the genes of groups VI and VII were almost all significantly upregulated in roots, stems and leaves after the heat treatment (Figure 7O-S). When looking at figure 7O, the gene Solyc03g115450.1.1 is not upregulated in leaves. Same thing for gene Solyc02g068980.1.1 which is not expressed at all in leaves at all times.
320. The authors claim that genes Solyc06g075780.1.1 and Solyc05g006310.2.1 were strongly upregulated in the roots and leaves. Figures 8I and 8K show that these 2 genes are not expressed at all in leaves.
323. Seven not six genes are upregulated in the roots, stems or leaves.
326-328. The expression patterns of the genes belonging to the same group were not similar in all the tissue, there was a lot of variability that is tissue-dependent. So the claim in these lines is not accurate
390. Should read: Among the C2H2-ZPs characterized, Zat6…
392. Should read ZAT10/STZ enhances (no “could).
393. Should read: OsZFP213 interacts with OsMAPK3 and enhances salt
395. Take out “were” and “the”
397. Take out “therefore” and “the” preceding 34 C2H2-ZF
413. Should read: we found that it was also induced by the heat treatment
417. Should read: heat treatment are interesting and should be explored in the future.
427. Should read: a strategy to improve heat tolerance of tomato plants.
Overall, the manuscript contains important information regarding newly identified C2H2-ZFPs in tomato. The phylogenetic and structural analysis were done adequately but the expression analysis need to be re-interpreted and re-written in the result and the discussion sections. The authors use this analysis to validate the expression data they retrieved for the database and speculate about the possible function of these genes in stress tolerance, so it is imperative that the data be interpreted accurately. Figure 3 also needs some revision to be more clear and easy to read.

Reviewer 4 ·

Basic reporting

It was interesting about investigation of C2H2 zinc-finger genes and their expression patterns in tomato leaf, root and stem under heat stress. This manuscript was written detailed overall. But, there is not described the major key point on manuscript. I fell that a style and contents of the manuscript are difficult for readers to know and/or understand what the authors would like be focusing in results and discussions. The authors should summarize the discussion section concisely and simply. Because mostly they were overlapped with the contents in results sections. Also, the authors missed figure legend of supplemental data. I recommend to revise the all content of the manuscript.

Experimental design

no comment

Validity of the findings

no comment

Additional comments

It was interesting about investigation of C2H2 zinc-finger genes and their expression patterns in tomato leaf, root and stem under heat stress. This manuscript was written detailed overall. But, there is not described the major key point on manuscript. I fell that a style and contents of the manuscript are difficult for readers to know and/or understand what the authors would like be focusing in results and discussions. The authors should summarize the discussion section concisely and simply. Because mostly they were overlapped with the contents in results sections. Also, the authors missed figure legend of supplemental data. I recommend to revise the all content of the manuscript.

Minor points and other comments
1. You should edit the alignment of text. All text is aligned with the left except for references.
2. I recommend summarize the examples of C2H2-ZFPs (Line 78-90) in introduction.
3. In Figure 1, It is necessary to get clearly grouping standard about homology. The proteins were categorized into nine groups based on their homology so why didn’t you divide group Ⅲ?
4. Add NJ and ML detail parameter description. (line 133-135)
5. Plz, describe the normalization method in more detail (line 154)
6. In Line 214 and 285, you should revise the references as same font to the paragraph.
7. Line 235-236 seven to 31. Is it 7 to 31?
8. Line 268, I think you missed the Solyc09g009030.3.1. Check the group Ⅱ.
9. You should be revise the image quality of Figure 3 that is hard to identify the gene name.
10. Solyc06g075780.1.1 and Solyc05g006310.2.1 are upregulated in roots and stems showed in Figure 8. In Line 319-320, the authors are described as these genes are upregulated in roots and leaves. Please check this point.

11. I think it would be better to add a legend about the Fig 5 color.

12. The authors should be explain sum up the key points about Figure 7 and 8 that would be more helpful the readers to interpret about the results.
13. In Figure 7, You should write about 18 selected C2H2-ZF genes criteria in the legend. What are those C2H2-ZF genes yardstick of selection?
14. Fig S2A, B and C have different explanations as to which method they use. (line 211-212 and line 227-229)

---

## Round 0.2 · Minor Revisions

Authors are suggested to make minor revision. Please take all comments into consideration and make revisions accordingly.

Reviewer 1 ·

Basic reporting

no comments.

Experimental design

no comments.

Validity of the findings

I still have a query regarding the grouping of all C2H2-ZFP sequences from Arabidopsis and tomato. they were grouped into six groups (A to F) in figure 1A, while tomato members were grouped into nine groups (I to IX). This provides a different grouping pattern. Uniform grouping pattern will provide a clear idea about the family diversification and differentiation.

Reviewer 2 ·

Basic reporting

The authors have done some work and responded to some of my concerns. However, there are still some problems remain to be solved.

Experimental design

The authors have resolved my concerns.

Validity of the findings

1.Tree:
It is still not clear how Figure 1A is largely consistent with Figure 1B. The authors said that some genes in Figure 1A are closely related to Figure 1B. However, I did not see any direct evidence from their description. They did not mention by what criteria they define ‘closely related’. Maybe they could try to pinpoint those genes in the figure. They claimed that this method has been widely used in Liu et al., 2015 and Yang et a., 2016. While Yang et al., is a chinese manuscript I could not access, Liu et al., did not perform any analysis regarding consistency between different trees. Please clarity on that.

2.Heatmap:
L280, it seems that group III is very similar to II and IV. Please explain why this is excluded. In addition, the x-axis label of heatmap is unclear. I am not sure which column refers to fruit development (L290). The authors may also consider pinpoint the genes discussed in the paper on the figure. It is very hard to locate them in the figure.

3.qPCR:
First, I would suggest the authors to revise the first sentence of this section on L295 (to confirm the reliability ….). It seems that you had talked about qPCR ahead of this. Second, the statement that ‘genes with similar expression patterns were classified into the same clusters’ seems to be reckless. Except for the last group, genes in other groups all have different expression patterns. Third, the authors need to rewrite the description of gene expression patterns and improve the organization. The current version is inaccurate. Many genes they claimed to increase/decrease at 24h were actually DE at 12h. Many genes they described as ‘no change’ were actually lowly expressed, beyond their range of detection. Furthermore, the description is not well-organized nor reader-friendly. The authors could focus on genes with distinct pattern and only illustrate a few examples. Meanwhile, it is a good idea to label the group of genes in Figure 7 and 8 so that the audience would have a better idea. Finally, the authors need to clarify what their t-test was compared to. I assume should be values at 0h.

4. Discussion:
The authors said ‘classification of C2H2-ZFs in NJ tree displayed the difference’. Isn’t it conflicting with their statement on consistency? In addition, the authors talked about similar functions regarding flowering on L425. Do those genes belong to the same clade in Figure 1A?

Additional comments

Minor points:
1.L65, petunia should not be italic.
2.Please clarify the sentence ‘indicate we could assume the putative function of these genes in tomato’ on L196. What kind of putative function?

---

## Round 0.3 · Minor Revisions

Please take comments from the reviewer#2 into consideration, and make changes accordingly.

Reviewer 2 ·

Basic reporting

The authors have solved some of my concerns. There are still a few points that need to be clarified.
First, for figure 1, they mentioned ‘some genes in Group VIII and IX was closely related to Group B’. However, there are many genes also related to group A based on the color code. It looks that Group VII is also related to group A. Meanwhile, there is a green clade that does not have correspondence in Figure 1B. Therefore, the claim that Figure 1A is consistent with Figure 1B may not stand.
Second, on L284 and 287, the authors stated that group VIII have both patterns of high expression and low expression?
Third, the authors did not explain what the textbox refers to in figure 6. Presumably genes in group IX showing similar patterns? I am very confused about what category of the genes with arrow correspond to. And they did not label all the genes discussed in the text. Did not find the labels of Solyc03g025440.3.1, Solyc04g056320.2.1, Solyc06g065440.1.1, and Solyc11g017140.2.1.

Experimental design

no comment

Validity of the findings

no comment

Additional comments

no comment

---

## Round 0.4 · Minor Revisions

Please make the final remaining corrections according to the suggestion from reviewers

Reviewer 2 ·

Basic reporting

The authors have addressed most of my concerns. There are two minor points that need to be modified.
1. L205, should be ‘clustered’.
2. L203, the authors’ claim that ‘some genes in Group VII, VIII and IX was closely related to Group A and B’ is not true, consider group VII is only related to group A not B.

Experimental design

No comment

Validity of the findings

No comment

Additional comments

No comment

Reviewer 3 ·

Basic reporting

The manuscript is written in scientifically appropriate English
The literature cited is ample and provides somewhat an adequate background on the paper’s topic. The figures are relevant, well labeled and well described. The authors addressed my concerns about figure 3. The gene annotation is now clear

Experimental design

No comments on the experimental design. It's adequate

Validity of the findings

The authors addressed my concerns regarding the interpretation of the gene expression data

Additional comments

I thank the authors for addressing my minor revisions. I have no revisions to add.

Reviewer 4 ·

Basic reporting

It was still interesting about investigation of C2H2 zinc-finger genes and their expression patterns in tomato leaf, root and stem under heat stress. This manuscript was written detailed overall. I have no doubt on author’s results. The authors have addressed the issues raised in my previous review.

I hope author could be clear about below (minor comment)
1. Figure 1 is very small to confirm the gene names as belong to groups. I think that listing of gene names in suppl. data additionally could be better to readers.
2. Authors should check or change the inequality sign as ≤ in P.11 Line 267.

Experimental design

'no comment'

Validity of the findings

'no comment'

---

## Round 0.5 · Minor Revisions

The Authors made changes on the manuscript according to the comments from reviewers.

#Staff Note - although the Editor is now happy that the article is Acceptable, Gerard Lazo, the Section Editor for this part of the journal has an additional request before final acceptance. His comments are:

"As 104 members of this family were analyzed and expression was measured over tissues and development an annotation method should be applied. Regarding the sequences for the peptides found in supplemental data are these tied to specific chromosome locations which can be annotated in some way. Likewise the genome version or release is not mentioned other than the genome database resource. This would be important to distinguish any updates done over time.

Journal manuscripts are often scanned by text-mining software that locates and extracts core data elements, like gene function. Adding standard ontology terms, such as the Gene Ontology (GO, geneontology.org) or others from the OBO foundry (obofoundry.org) can enhance the recognition of your contribution and description. This will also make human curation of literature easier and more accurate. None of this was visible. I will leave this manuscript in a "minor revision" mode until the issues of genome version and annotations can be accomplished.”

---

## Round 0.6 · accepted · Accept

Authors have taken the comments into consideration, and made changes accordingly.